# Single-Cell Gene Network Analysis and Transcriptional Landscape of MYCN-Amplified Neuroblastoma Cell Lines

**DOI:** 10.3390/biom11020177

**Published:** 2021-01-28

**Authors:** Daniele Mercatelli, Nicola Balboni, Alessandro Palma, Emanuela Aleo, Pietro Paolo Sanna, Giovanni Perini, Federico Manuel Giorgi

**Affiliations:** 1Department of Pharmacy and Biotechnology, University of Bologna, 40138 Bologna, Italy; daniele.mercatelli2@unibo.it (D.M.); nicola.balboni@unibo.it (N.B.); alessandro.palma3@studio.unibo.it (A.P.); 2IGA Technology Services, 33100 Udine, Italy; ealeo@igatechnology.com; 3Department of Immunology and Microbiology, The Scripps Research Institute, La Jolla, CA 92037, USA; psanna@scripps.edu

**Keywords:** neuroblastoma, gene networks, single-cell, transcriptomics, master regulator analysis

## Abstract

Neuroblastoma (NBL) is a pediatric cancer responsible for more than 15% of cancer deaths in children, with 800 new cases each year in the United States alone. Genomic amplification of the MYC oncogene family member MYCN characterizes a subset of high-risk pediatric neuroblastomas. Several cellular models have been implemented to study this disease over the years. Two of these, SK-N-BE-2-C (BE2C) and Kelly, are amongst the most used worldwide as models of MYCN-Amplified human NBL. Here, we provide a transcriptome-wide quantitative measurement of gene expression and transcriptional network activity in BE2C and Kelly cell lines at an unprecedented single-cell resolution. We obtained 1105 Kelly and 962 BE2C unsynchronized cells, with an average number of mapped reads/cell of roughly 38,000. The single-cell data recapitulate gene expression signatures previously generated from bulk RNA-Seq. We highlight low variance for commonly used housekeeping genes between different cells (ACTB, B2M and GAPDH), while showing higher than expected variance for metallothionein transcripts in Kelly cells. The high number of samples, despite the relatively low read coverage of single cells, allowed for robust pathway enrichment analysis and master regulator analysis (MRA), both of which highlight the more mesenchymal nature of BE2C cells as compared to Kelly cells, and the upregulation of TWIST1 and DNAJC1 transcriptional networks. We further defined master regulators at the single cell level and showed that MYCN is not constantly active or expressed within Kelly and BE2C cells, independently of cell cycle phase. The dataset, alongside a detailed and commented programming protocol to analyze it, is fully shared and reusable.

## 1. Introduction

For decades, cell lines have been widely used in cancer biology as a standard setting to investigate molecular mechanisms and to test the effects of genetic and chemical perturbations. Experiments performed on cell lines often lay the foundation for further investigation of biological response in animal models, ultimately providing valuable translational inputs for clinical medicine [1]. After the advent of the omics era, scientists gained the capability to match specific cell lines to the genomics, epigenomics and transcriptomics features of specific tumor subtypes [2]. In recent years, advances in sequencing-based diagnostics have allowed researchers to choose, virtually in real time, the top matching cell line model for individual cancer patients as a key step of precision medicine approaches [3].

One of the main advantages of cell line-based experiments is their high reproducibility [4], based on the fact that cell lines are genetically stable and transcriptionally more homogeneous than in vivo tumor models [5]. However, it has been shown that cell lines from different labs may show genetic and transcriptional alterations, due to clonal and evolutionary divergences, which may account for occasional differences in phenotypes and pharmacological responses [6]. While the between-lab diversity of cell lines has been critically and widely recognized [7], the within-lab and within-plate heterogeneity of cell line cultures is often ignored, despite evidence that spatial biases exist even in cell cultures [8].

Recent technological advances have provided the unprecedented opportunity to investigate heterogeneity at single-cell resolution [9], allowing researchers to quantitatively identify cellular subpopulations and to uncover molecular mechanisms underlying phenotypic diversity among cells [10]. The majority of the single-cell RNA-Sequencing (scRNA-Seq) studies published so far in cancer biology aim at deciphering tumor tissue samples’ composition complexity and the interplay between cancer cells and the cellular components of the tumor microenvironment [11]. ScRNA-Seq has been successful in delineating the previously uncharacterized histological heterogeneity of many different tumors [12,13,14]. In cell lines, single-cell sequencing has been applied to investigating the early insurgence of drug resistance mechanisms [15,16] and tumor evolution [17,18].

Among tumors, Neuroblastoma (NBL) is a representative example of a highly histologically heterogeneous cancer [19]. NBL is the most common extracranial solid tumor of early childhood arising from neural crest cells, showing a wide spectrum of clinical behavior, spanning from spontaneous regression without chemotherapy to a frequent metastatic manifestation with a drug-resistant phenotype, especially in older patients [20,21,22]. There exist at least three different molecular subtypes in which aggressive NBLs can be categorized, named Mesenchymal, 11q Loss of Heterozygosity and MYCN-Amplified, the last two of which are characterized by specific genomic alterations [23]. The MYCN-Amplified subtype comprises roughly 20% of all NBLs, and 50% of high-risk patients, constituting the most aggressive and least treatable form of this cancer [24]. There exist several cell lines derived from MYCN-Amplified NBLs, which have been widely characterized both transcriptionally by RNA-Sequencing [25], and epigenetically by ChIP-Sequencing [26,27,28] and ATAC-Sequencing [29].

Despite these considerable characterization efforts, all current sequencing of MYCN-Amplified cells has been performed on bulk samples, and therefore constitutes an average of all cells within one or a few culture dishes.

In the present work, we provide the first scRNA-Seq dataset of two of the most used MYCN-Amplified NBL model cell lines, namely SK-N-BE (2)C (herein referred to as “BE2C” for brevity) and Kelly. We used a combination of 10× Genomics technology and Illumina to carry out scRNA sequencing of two unsynchronized cell cultures. We extracted 962 BE2C cells and 1105 Kelly cells with an average number of genome-mapped reads/cells of 38,334 for BE2C and 37,760 for Kelly. The dataset is provided both as raw sequencing reads (provided at the Sequence Read Archive as BAM files aligned on human genome version hg19, but also containing unaligned reads) and as processed gene counts matrices in the Appendix A.

Each cell was investigated with respect to individual gene expression (with a focus on commonly used housekeeping genes), single-cell pathway enrichment analysis and single-cell master regulator analysis (MRA) [30]. The manuscript is accompanied by a detailed executable R markdown document recreating all steps of the analysis and providing researchers with a standard pipeline of investigation, using state of the art R packages for normalization, summarization and plotting. Bringing single-cell resolution to transcriptome quantification of cell lines used to investigate the deadly NBL pediatric cancer will allow researchers to better appreciate the heterogeneity of cells in seemingly homogeneous cell culture settings. We believe our study adds a fundamental layer of transcriptomics complexity that cannot be extracted from classic bulk sequencing datasets.

## 2. Materials and Methods

### 2.1. Cell Cultures

Cell lines were obtained from Sigma-Aldrich^®^. BE2C cells were cultured in high glucose DMEM (Sigma-Aldrich^®^, St. Louis, MO, USA) +10% fetal calf serum (FCS, Gibco^®^); Kelly cells were cultured in RPMI 1640 (Sigma-Aldrich^®^) +10% FCS (Gibco^®^). Both media were supplemented with 2 mM L-Glutamine and 1% Penicillin/Streptomycin. BE2C and Kelly cells were processed separately in all steps of cell culturing. Cells were grown adherently in standard T-25 flasks at 37 °C with 5.0% CO_2_ [31] and passaged by trypsinization at ~75% confluency. A flask each of BE2C and Kelly cells at 70% confluency (Figure 1A for BE2C, Figure 1B for Kelly) was filled to capacity (roughly 83 mL of volume) with growth medium at 37 °C and tightly sealed for transport to the sequencing facility (roughly 25 min away) for sequencing with the Chromium^®^ instrument for 10× Genomics^®^ (Pleasanton, CA, USA) library preparation. No cell cycle synchronization strategy was used during the cell culture steps.

### 2.2. 10× Genomics Library Preparation and Sequencing

Cells were harvested using trypsin-EDTA solution and centrifuged. The pellet was resuspended in PBS 1× containing bovine serum albumin (BSA) 0.04%. Cell concentration was determined using the Countess II FL Automated Cell Counter (Thermo Fisher Scientific^®^ Waltham, MA, USA). Trypan Blue staining was used to assess cell viability. Chromium controller and Chromium Single Cell 3′ Reagents Kit v2 (10× Genomics^®^) were used for partitioning cells into gel beads-in-emulsion (GEMs), where all generated cDNA shares a common 10× barcode. Libraries were generated from the cDNA and checked with both Qubit 2.0 Fluorometer (Invitrogen^®^, Waltham, MA, USA) and Agilent Bioanalyzer DNA assay (Agilent^®^, Santa Clara, CA, USA). Libraries were prepared for sequencing following manufacturer’s instructions (Illumina^®^, San Diego, CA, USA) and then sequenced in 150 bp paired-end mode on Illumina^®^ HiSeq2500.

### 2.3. Data Processing

Raw reads were mapped to the human genome version hg19/GRCh37 using the STAR aligner version 2.7 [32]. The resulting aligned reads data were saved in BAM format, which also included unaligned reads. This BAM was then processed with Cell Ranger v4.0.0, in order to obtain matrices of gene counts per cell in CSV (comma-separated value) format.

Gene count matrices were loaded in the R statistical software version 4.0.2, running Bioconductor version 3.11. Plotting was performed using base R functions and the corto package version 1.1 [33]. Raw gene counts were normalized using the transcripts per kilobase million (TPM) method. Briefly, in TPM normalization, gene-wise read counts were divided by the length of each gene (defined by the UCSC database) in kilobases. These values, cell by cell, were then divided by the number of reads (in millions) mapped in each cell. Correlation values were calculated using Spearman’s method. For data dimensionality reduction and clustering analysis, normalization was performed on raw gene counts with the Seurat package version 3.2.1 [34] using the LogNormalize method with a scale factor of 10,000. Clustering was also performed using the Seurat package after removing genes measured in less than 3 cells (out of 962 BE2C cells and 1105 Kelly cells, for a total of 2067 cells). Assignment of cells to cell cycle phases was performed using the Seurat package with cell cycle genes defined by the Regev and Garraway labs [35]. The variance shown in Figure 2B is the residual variance after subtracting the expression levels using a loess regression (otherwise, the expression variance would always be highly correlated to average expression).

Pathway enrichment analysis was performed using gene set enrichment analysis (GSEA) as described before [36] using pathway definitions from MSigDB [37], KEGG [38] and Reactome [39] databases. MRA and GSEA single-cell analyses were performed using functions from the R suite corto [33] as described in the Appendix A. The normalized enrichment score (NES) calculated by corto indicates the magnitude of up- or down-regulation of the TF network (i.e., the collection of targets and their weights [40]) and is calculated as the enrichment score of the corto analysis (applying the network on the signature, in our case the BE2C vs. Kelly comparison) divided by the mean enrichment score of all permutations (calculated by shuffling both networks and samples 1000 times). A Benjamini–Hochberg-corrected *p*-value [41] is linearly associated with the NES, and specifies the expected occurrence of permuted networks with an enrichment score greater than or equal to the observed one.

The Harenza dataset [25] was used to compare the scRNA-Seq data with bulk RNA-Seq. The dataset was downloaded from Gene Expression Omnibus (entry GSE89413). Bulk and sc datasets were TPM-normalized to make them comparable.

## 3. Results

### 3.1. Characterization of Landmark Gene Expression

Under optical microscopy inspection, the appearance of both BE2C cells (Figure 1A) and Kelly cells (Figure 1B) at ~70% confluence was consistent with the previous literature [42,43]. We quantitatively checked the presence of several genes, in terms of average expression across the entire cell population (expressed as Log10 average TPM) and in terms of number of cells with at least one read mapped on the gene (Figure 1C for BE2C, Figure 1D for Kelly). We checked the expression of four commonly used housekeeping genes: ACTB, GAPDH, B2M and GUSB [44]. All these genes are highly expressed and could be detected in almost all cells, with the exception of GUSB, detected in only ~50% of both Kelly and BE2C cells. As expected for MYCN-Amplified cells, the MYCN gene is also amongst the most expressed, both in absolute TPMs and as number of expressing cells. Its paralogs, MYCL and MYC, are expressed in extremely low amounts, in only a few cells. As shown before [25], MYCN has a slightly higher expression value in BE2C cells (Figure 1E) compared to Kelly cells (Figure 1F). We also confirmed the higher expression, in Kelly as compared to BE2C, of the NBL oncogene LMO1, as shown before [45]. The ALK gene, which carries a F1174L mutation in Kelly and is WT in BE2C [25], is expressed at low levels in Kelly and is barely detectable in BE2C. Among the most expressed genes in both cell lines are those encoding for ribosomal proteins, such as RPL37 and RPS9. Amongst crucial factors of the MYCN regulatory network, including PRDM8, MYBL2, HMGB2 and TEAD4 [23], HMGB2 showed the highest and most robust mRNA levels.

Kelly cells are characterized by high average levels of metallothionein genes, such as MT2A, MT1X and MT1E, which are, however, detected in only a fraction of cells and therefore display a high expression variance. The expression level of metallothionein genes has been correlated with intracellular levels of metal ions (e.g., MT1X for copper [46]) and their expression variance represents the most notable difference when compared between BE2C and Kelly cells (Figure 2A,B). The two cell lines possess highly similar expression profiles (Spearman Correlation Coefficient, SCC = 0.883) in terms of average expression, with genes such as GAPDH, ACTB and MYCN highly expressed in both, with very low expression of MYC and MYCL (Figure 2A). The two cell lines are highly similar when comparing gene expression variances (Figure 2B, SCC = 0.781), where metallothionein genes are the ones most characterizing the divergence between the two, with a much higher variance in Kelly cells.

### 3.2. Comparison with Bulk RNA-Seq

The two single-cell datasets recapitulate the information contained in bulk data generated by another study [25]. We summed the gene TPMs across all single cells and correlated these values with TPMs from bulk RNA-Seq (Figure 2C for scBE2C vs. bulk BE2C, and Figure 2D for scKelly vs. bulk Kelly). The overall expression is highly correlated (SCC = 0.85 for BE2C, SCC = 0.91 for Kelly), showing that single-cell sequencing is capable of recreating a bulk experiment, adding extra information from individual cells. A TSNE visualization of all the bulk RNA-Seq from the Harenza NBL cell lines dataset shows that the profile most similar to scKelly is bulk Kelly cells (Figure 2E). On the other hand, our BE2C single-cell dataset is most correlated with both BE2C and BE2 (SK-N-BE-2, from which BE2C derive), according to both TSNE visualization (Figure 2E) and whole-transcriptome expression Spearman Correlation Coefficient analysis (Figure 2F). See also the attached Appendix A, section “Comparison with bulk RNA-Seq data”, for a full comparison with existing NBL cell lines sequenced at bulk resolution [25].

### 3.3. Dimensionality Reduction and Clustering of Cells

When clustered together, BE2C and Kelly cells show very distinct properties, being highly separated by both UMAP (Uniform Manifold Approximation and Projection, Figure 3A) and TSNE (t-distributed Stochastic Neighbor Embedding, Figure 3B) projections. The Louvain method [47] shows two main clusters, clearly separating Kelly and BE2C cells. However, increasing the resolution parameter highlights two subpopulations for BE2C cells (Figure 3A, see also Appendix A, section “Louvain clustering”, for more details). The 20 marker genes most different between BE2C cluster 2 vs. BE2C cluster 1 are shown in Table 1. Among these, we observed many genes coding for ribosomal proteins, such as RPSA, RPL35A and RPL15, but also VCAN, which is expressed in 27% of BE2C cluster 2 cells, and in only 26% of BE2C cluster 1 cells. VCAN codes for the versican protein, a structural component of the extra cellular matrix in brain cells, and is considered to be a pro-inflammatory driver of tumor progression [48].

Being unsynchronized, both cell populations appear to be in different cell cycle phases (Figure 3C), with Kelly cells appearing predominantly in S phase (57.47%) and BE2C cells more evenly distributed between G1, S and G2/M phases. More BE2C cells appear to be in G1 phase (28.69%) than Kelly cells (19.73%). It has been shown elsewhere, in embryonic stem cells, that more undifferentiated cells tend to spend a larger proportion of the cell cycle in S phase, with shortened G1 and G2 phases [49]. The observed distributions of BE2C and Kelly do not seem to correlate with known proliferation parameters of the two cell lines: according to ATCC^®^ [50], the doubling time of BE2C cells is roughly 18 h, while according to the ExPASy database [51], the doubling time for Kelly cells is roughly 30 h.

The cell cycle is a major component of the observed TSNE-reduced structure of the cell lines (Figure 3D). Another observable major source of variability is the number of measured mapped reads per cell (Figure 3E). Globally, the cells in our dataset were measured with a mean number of mapped reads of roughly 38,000 (38,334.42 for Kelly and 37,760.29 for BE2C), with most of the cells having roughly 30,000 reads and only a handful of cells surpassing the 100,000 reads threshold (Figure 3E).

### 3.4. Heterogeneity of Gene Expression

Our dataset can be used to detect the heterogeneity of expression of specific genes within the cell populations, in terms of Log10 TPM (Figure 4). The housekeeping ACTB gene is more expressed in BE2C cells (Figure 4, cluster above), and ranges within one order of magnitude of expression (roughly 630-9772 in non-logarithmic scale TPM). Similar considerations can be applied to the other two housekeeping genes, B2M and GAPDH. ALK displayed low expression levels in the majority of the dataset, while both LMO1 and MYCN show notable differences across the dataset. Overall, this dataset shows an unprecedented variability of gene expression within MYCN-Amplified cell lines, which supports further investigation of cancer cell line models via single-cell sequencing.

### 3.5. Differential Gene Expression

We aimed at characterizing the differences between BE2C and Kelly cells using our dataset, comparing 962 BE2C cells vs. 1105 Kelly cells with the Seurat pipeline. Our analysis shows a positive correlation with the bulk RNA-Seq BE2C vs. Kelly, with a correlation of 0.39 based on transcriptome-wide log2FC (see Appendix A, “comparison with bulk signature” paragraph). The differences are marked, with 7645 genes upregulated in BE2C vs. Kelly cells and 3099 downregulated, at a significance threshold set at adjusted *p*-value = 0.01 (adjusted by the Benjamini–Hochberg method [41]). This high number of differentially expressed genes, corresponding to roughly half of the transcriptome, suggests that the number of samples is allowing the statistical tests to deem significant even small changes with log2FC < 0.1. The number of significant genes drops to 3254 upregulated/1104 downregulated in BE2C at an adjusted *p*-value threshold of 10^−20^ and 622 upregulated/257 downregulated at an adjusted *p*-value of 10^−100^. The most upregulated gene in BE2C cells (when compared to Kelly) is RPS25, coding for a ribosomal protein, as is the most upregulated gene in Kelly, RPL27: as indicated in the next section, there are marked differences in how the two cell lines express ribosomal genes and pathways. MYCN is more expressed in BE2C than in Kelly (adjusted *p*-value = 4.70 × 10^−44^), probably due to the higher copy number of the MYCN region in BE2C cells [25]. See Appendix A “Visualization of differential expression by volcano plot” paragraph and associated table for the full analysis.

### 3.6. Pathway Analysis

We analyzed pathway enrichment both as a comparison between BE2C cells and Kelly cells, and within each cell (Figure 5). The overall analysis highlights that BE2C cells have a markedly higher expression of genes associated to Epithelial-Mesenchymal Transition (EMT) (Figure 5A,B), a pathway generally associated with higher proliferation, chance of metastasis, poor survival, and drug resistance [52]: it can be hypothesized therefore that BE2C cells are a better model for highly aggressive MYCN-amplified NBL than Kelly cells. Another strongly upregulated BE2C-specific pathway is the signaling downstream of EGFRvIII, a mutated version of EGFR lacking ligand binding domain, often amplified in tumors [53] (Figure 5B). On the other hand, as discussed in the previous section, Kelly and BE2C differ in the expression of ribosomal protein-coding genes (Figure 5A): a marked upregulation of rRNA metabolism and protein translation was observed in Kelly cells (Figure 5A,B). Kelly and BE2C cells differ dramatically in the Reactome-defined pathway “Nervous System Development” (Figure 5B) [39], which is upregulated in Kelly, indicating a higher differentiation of these cells compared to BE2C, which is supported by the higher mesenchymal pattern of BE2C cells (Figure 5B), according to GSEA profiling.

We then analyzed the levels of relative pathway expression at the single-cell level [54,55], providing a cell-by-cell analysis of all the 24,472 pathways from the Molecular signatures database (MsigDB [37]) collection (available on the R markdown paragraph “Single-cell GSEA” and associated results). As observed before, the ribosome-associated genes are collectively upregulated in Kelly cells (Figure 5C, bottom group; see also Figure 3B for reference assignment of cell types), but show a noticeable variance in BE2C cells: in these cells, ribosome-associated protein-coding genes appear upregulated in cells in G1 phase (compare Figure 3D and Figure 5C). There are also heterogeneities within cells from the same culture dish that are not attributable to cell cycle differences. For instance, an NBL-related important pathway, the “Hallmark MYC canonical targets” in the MsigDB collections (Figure 5C, bottom), shows high heterogeneity within both BE2C and Kelly populations, without a clear association with cell cycle phase (Figure 3D).

### 3.7. Master Regulator Analysis

Master regulator analysis (MRA) aims at defining key transcription factors which are likely to control the observed transcriptional changes in a specific perturbation or comparison [33,56,57]. This analysis can be performed between groups of samples (e.g., in our case, all BE2C vs. all Kelly cells) or on a sample-by-sample basis [56]. Such an analysis requires the transcriptome-wide definition of gene networks [57], often based on coexpression analysis [58]. In this dataset, we used two networks commonly used in Neuroblastoma research, based on data from the TARGET (Therapeutically Applicable Research To Generate Effective Treatments) and NRC-Siopen consortia [23], and a network generated from the Kocak Neuroblastoma cohort [59] via the corto R package [33]. Using these networks, we performed a full MRA via the corto package, in order to highlight differential activity of transcription factors in BE2C vs. Kelly cells. The results appear to be robust, showing a high agreement when using different datasets to generate network models (Figure 6A). The common master regulators identified when interrogating independent networks are: DNAJC1, ETV4, HEYL, HINFP, MBD3, NFRKB, NPAT, SCYL1, TAF10, TAF6, TWIST1, ZCCH24, ZNF25, ZNHIT1 (all upregulated in BE2C cells) and SESN2, TRIM28, UXT, ZNF581 (all upregulated in Kelly cells). Enrichment profiles of the networks of these transcription factors are shown using the NRC network (Figure 6B), the TARGET network (Figure 6C) and the Kocak network (Figure 6D).

Some of these differences are of notable relevance to NBL pathogenesis: one example is SESN2, upregulated in Kelly cells, a regulator of mTORC1. High levels of SESN2 are associated with apoptosis, while low levels are associated with drug resistance [60,61]. Another example is TWIST1, upregulated in BE2C cells, a direct coeffector of the MYCN pathway in NBL [62]. Other transcription factors are associated with cancer-related pathways, such as NPAT [63], ZNF264 [64], HEYL [65] and ETV4 [66].

The overall MRA of the BE2C vs. Kelly comparison hides, however, the heterogeneity of TF network activation within single cells. For example, SESN23 appears to be highly active only in a fraction of Kelly cells, as is ZNF264 (Figure 7). Amongst the TFs with the highest variance within cell types we find MAX, a well-known functional interactor of MYC and MYCN [67], but also the already cited ZNF264, together with other less-characterized zinc finger transcription factors ZNF429 and ZBTB43 (Figure 7).

Another strategy to investigate sources of heterogeneity in single-cell datasets is the single-cell latent variable model (scLVM) [68], which allows the identification of interpretable and non-interpretable sources of variability. We applied the latest implementation of the method, f-scLVM [69], in order to highlight what drives and explains the differences in transcriptome we observe in the Kelly/BE2C single-cell dataset (Figure 8A). We used gene annotations deposited in WikiPathways [70] to define annotated terms of heterogeneity. The two top terms associated with dataset heterogeneity are unannotated, or “hidden” sources of variability, and correspond to the observed differences between Kelly and BE2C cells and between the two BE2C major populations (Figure 3A). The genes most associated to the Kelly/BE2C variability are the ENG glycoprotein, a component of the TGFBR complex, and the transcription factor GATA4 (Figure 8B). The third source of heterogeneity can be mapped over the variability of genes associated with cholesterol metabolism (Figure 8C), like the 3-Hydroxy-3-Methylglutaryl-CoA Synthase 1 (HMGCS1) and the Methylsterol Monooxygenase 1 (MSMO1). The fourth term is cell cycle, which, as shown before (Figure 3D) is a strong component in determining the between-cell transcriptional differences of cultured neuroblastoma cells, as it is to be expected from unsynchronized cell cultures. The genes most involved in cell-cycle-specific heterogeneity are the driver of G2/M transition, PLK1, as well as the centrosome protein CENP2 and several cyclins (CCNB1, CCNB2 and CCNA2).

## 4. Discussion

We investigated by single-cell technology the transcriptome landscape of the two most used cell line models of MYCN-amplified NBL (Kelly and BE2C) at an unprecedented resolution. We confirmed that the most used housekeeping genes (B2M, GAPDH, ACTB) are characterized by both high expression and low variance in both cell lines. Metallothionein transcripts, while highly expressed, proved highly variable in Kelly cells. Our analysis shows that single-cell RNA-Seq data, when summing together all cell transcriptional abundances, is very similar to bulk RNA-Seq data, in this case generated by another lab [25], so much so that it is possible to clearly identify the dataset cell type based on a simple correlation analysis with the entire collection of NBL cell lines. Our analysis shows that the two cell lines are clearly transcriptionally distinct (Figure 3). Clustering analysis with higher resolution parameters highlights the presence of two BE2C subpopulations, which do not seem to be associated with common sources of variance, such as cell cycle or read coverage (Figure 3). Indeed, the expression of some key NBL genes, such as MYCN and LMO1, is not constant across the dataset, and some cells appear to have a surprisingly low expression of both (Figure 4). LMO1 is more expressed in Kelly cells, and this is compatible with previous literature [71]. In fact, Kelly cells have a different genotype at locus rs2168101 within LMO1 first intron, which is G/–in Kelly and T/– in BE2C. The Kelly G allele forms a GATA binding site, recruiting a transcriptional complex which increases levels of LMO1. BE2C cells do not possess this strong enhancer site, leading to a very low LMO1 detection.

While BE2C and Kelly cells are widely used as interchangeable experimental models for MYCN-amplified NBL, they showed transcriptional differences between them as well as variability within each cell line. The transcriptional differences highlighted here could be the basis for some observed experimental differences between the two cell lines, e.g., in the transcriptional machinery following glutamine deprivation, which induces apoptosis in BE2C cells, but apparently not in Kelly cells [72]. Our analysis suggests a more aggressive phenotype of BE2C cells when compared to Kelly cells. In fact, while BE2C cells appear to be more mesenchymal and with higher levels of MYCN (commonly associated with poorer survival in patients), Kelly cells appear on the whole to be more differentiated (Figure 5). However, both cell lines are to be considered as models of highly aggressive, stage 4 NBL [73], and Kelly cells are often considered a better model for cell migration and metastasis than BE2C [71].

Our study, beyond generating and analyzing this novel dataset, also extends the commonly used pathway enrichment and master regulator pipelines to single-cell analysis. We believe the current GSEA and MRA family of algorithms are optimally suited for single-cell data, despite the low coverage of individual cells, since the intrinsic noise of this measurement is diluted by aggregating many transcript levels into a single pathway or transcriptional network [74]. The results we obtained with MRA are robust, as they correlate well when using three different network models (Figure 6A).

In conclusion, we believe that single-cell RNA-Seq is able to fully recapitulate the biological findings of bulk RNA-Seq, and define further avenues of research for testing by delineating the cell-by-cell heterogeneity of individual genes, pathways and transcriptional networks. We believe our analysis to be entirely generalizable for other cell line studies, and we provide our entire analysis in a fully documented and reproducible R markdown document.

## Figures and Tables

**Figure 1 biomolecules-11-00177-f001:**
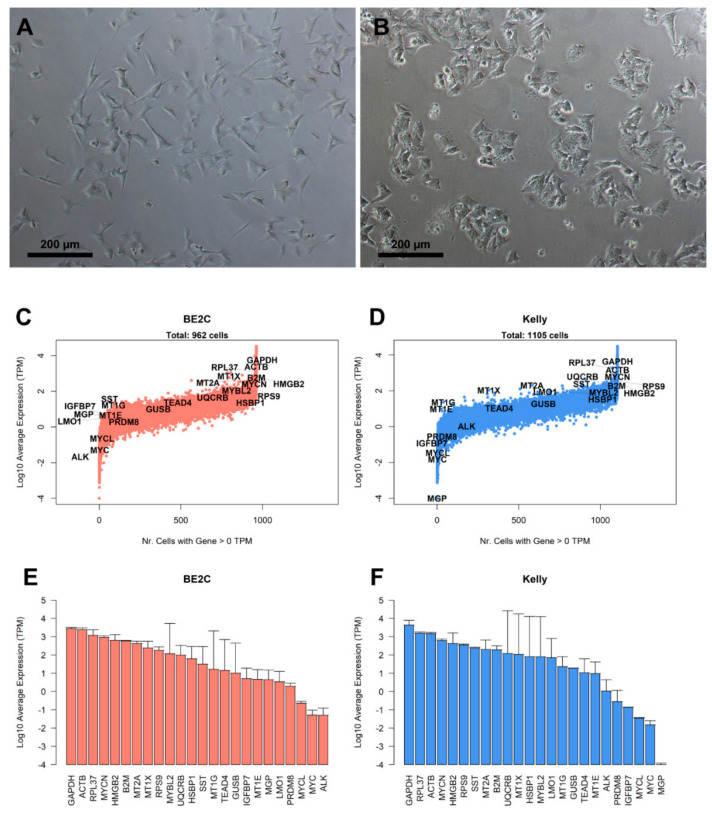
Initial analysis of single-cell expression on Kelly and BE2C cell lines. (**A**) BE2C and (**B**) Kelly cells prior to library preparation sequencing. (**C**,**D**) Plot showing the Log10 TPM average expression for all genes in the dataset (*y*-axis) and the number of cells where the gene is detected (*x*-axis) with TPM > 0 (i.e., more than one read) in BE2C and Kelly cells. (**E**,**F**) Selected representative genes shown as bar plots of overall log10 TPM average expression, with error bars depicting standard deviation in the dataset for BE2C and Kelly cells. A pseudovalue of 0.0001 (10^−4^) is added to TPM values (also in the next figures) prior to calculation of logarithm.

**Figure 2 biomolecules-11-00177-f002:**
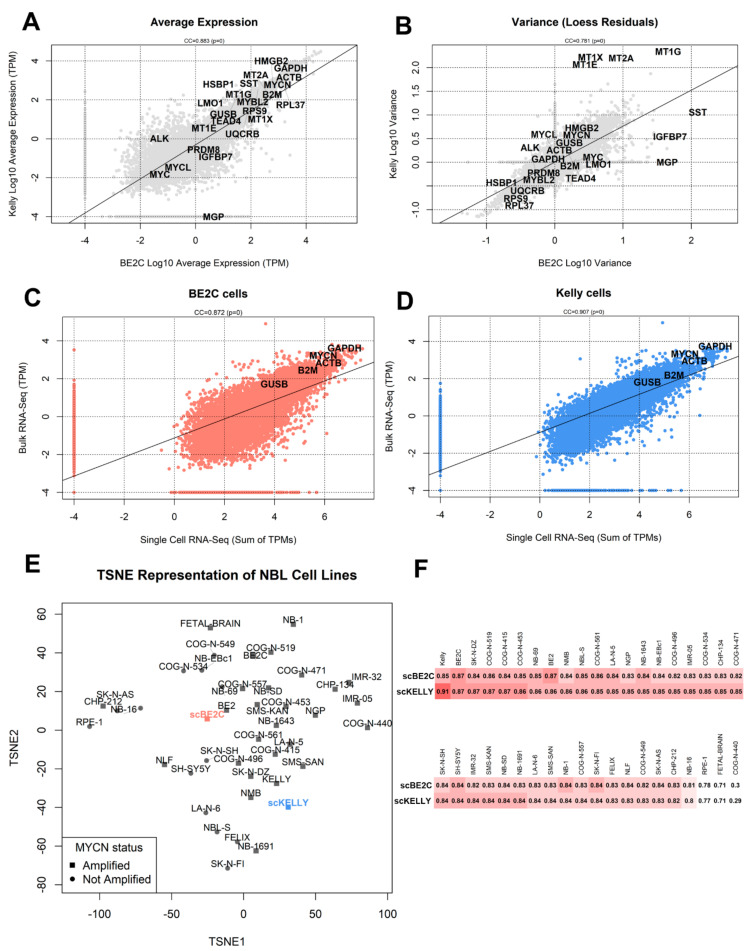
Comparison between BE2C and Kelly datasets and with bulk RNA-Seq. (**A**) Gene-by-gene comparison of log10 TPM average expression in BE2C (*x*-axis) and Kelly cells (*y*-axis). A linear regression line is shown, and the SCC is indicated with the correlation *p*-value (precision limit: 10-302). (**B**) Gene-by-gene comparison of log10 TPM variance of expression (after regressing out average expression with a loess regression) in BE2C cells (*x*-axis) and Kelly cells (*y*-axis). (**C**,**D**) Gene-by-gene comparison between single-cell dataset expression (*x*-axis, shown as log10 sum of TPMs) and bulk expression (*y*-axis, as log10 TPM expression of the cell line in the Harenza dataset [24]). The Spearman Correlation Coefficient (CC) is indicated. (**E**) TSNE visualization (calculated on TPM data) including the entire Harenza bulk RNA-Seq dataset [24] and the sum of TPMs of single-cell datasets. MYCN-amplified NBL cell lines are depicted as squares, not-MYCN-amplified cell lines as circles. (**F**) Heatmap reporting Spearman Correlation Coefficient between single-cell aggregated TPM data and 20 bulk RNA-Seq samples from the Harenza dataset. Samples are ordered by correlation coefficient with the scKelly sample, and reported in two rows for graphical convenience.

**Figure 3 biomolecules-11-00177-f003:**
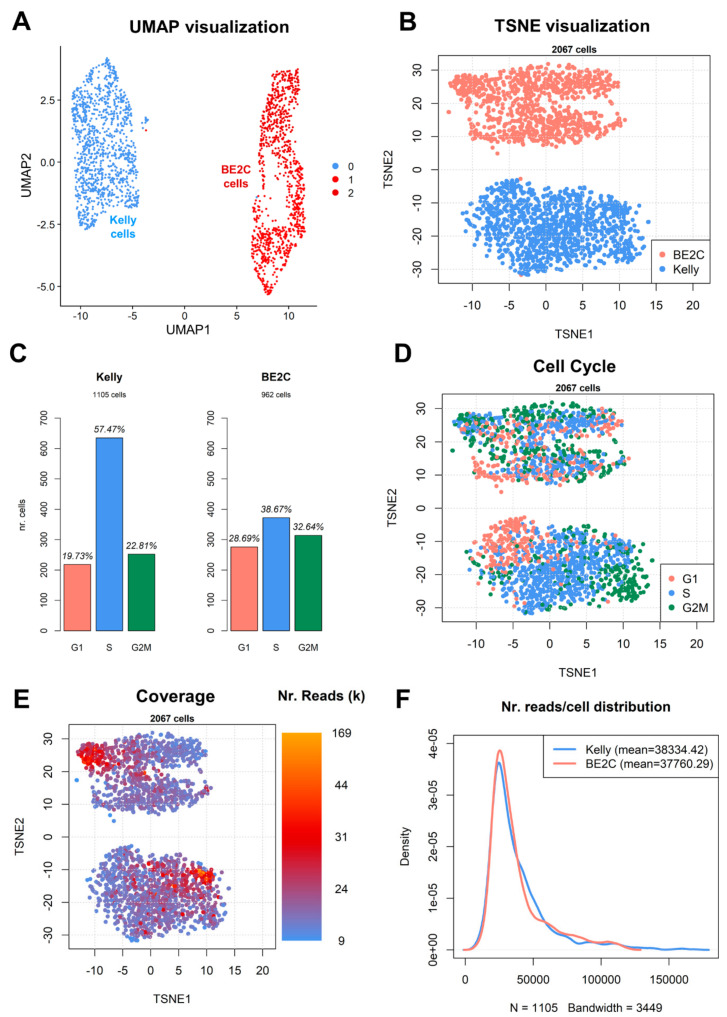
Visualization of single cells following dimensionality reduction. (**A**) UMAP and (**B**) TSNE representations of BE2C (red) and Kelly (blue) cells. Clustering assignment according to the Louvain method (high resolution parameters) is indicated (BE2C cells are divided into light and dark red). The numbers in panel A (0, 1, 2) correspond to inferred clusters. (**C**) Distribution of cells by predicted cell cycle phase. (**D**) Overlay of cell cycle phase over coordinates from panel B. (**E**) Overlay of nr of mapped reads (in thousands) per cell over coordinates from panel B. (**F**) Distribution of mapped reads/cell across the two single-cell datasets; *x*-axis: number of reads, *y*-axis: relative abundance of cells.

**Figure 4 biomolecules-11-00177-f004:**
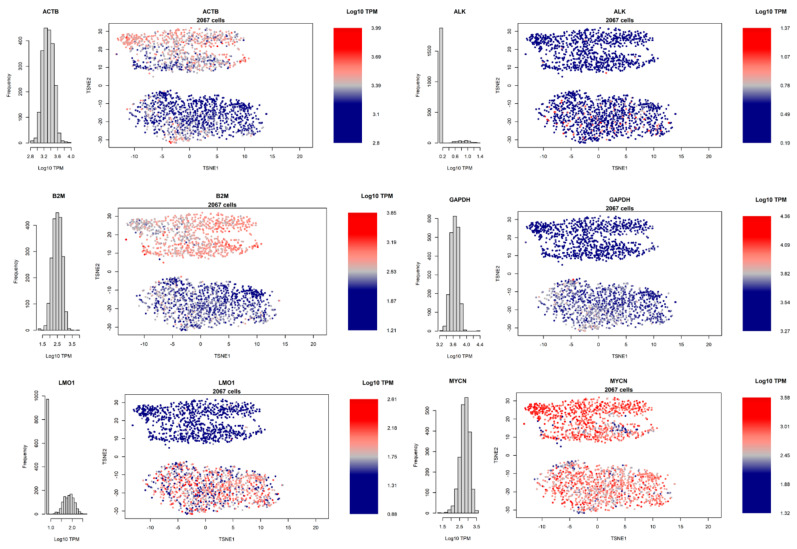
Single-cell distribution of selected genes, shown as log10 TPM. Color scaling is independent for each panel. Cartesian coordinates representing single cells are the same as Figure 3B.

**Figure 5 biomolecules-11-00177-f005:**
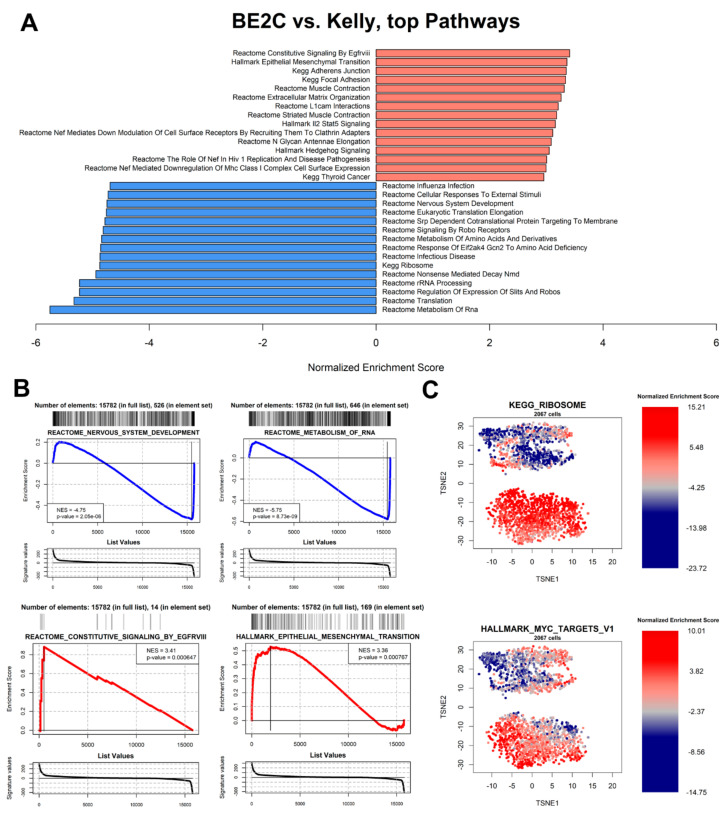
Pathway enrichment analysis. (**A**) Top ten upregulated and top ten downregulated pathways in the BE2C vs. Kelly cells comparison. The score is calculated using gene set enrichment analysis (GSEA) [36] as normalized enrichment score (NES). (**B**) Individual GSEA running score plots of four selected pathways in the BE2C vs. Kelly comparison. (**C**) Single-cell-specific NES of two selected pathways in the dataset. BE2C cells are on top, following the same cartesian coordinates as Figure 3B.

**Figure 6 biomolecules-11-00177-f006:**
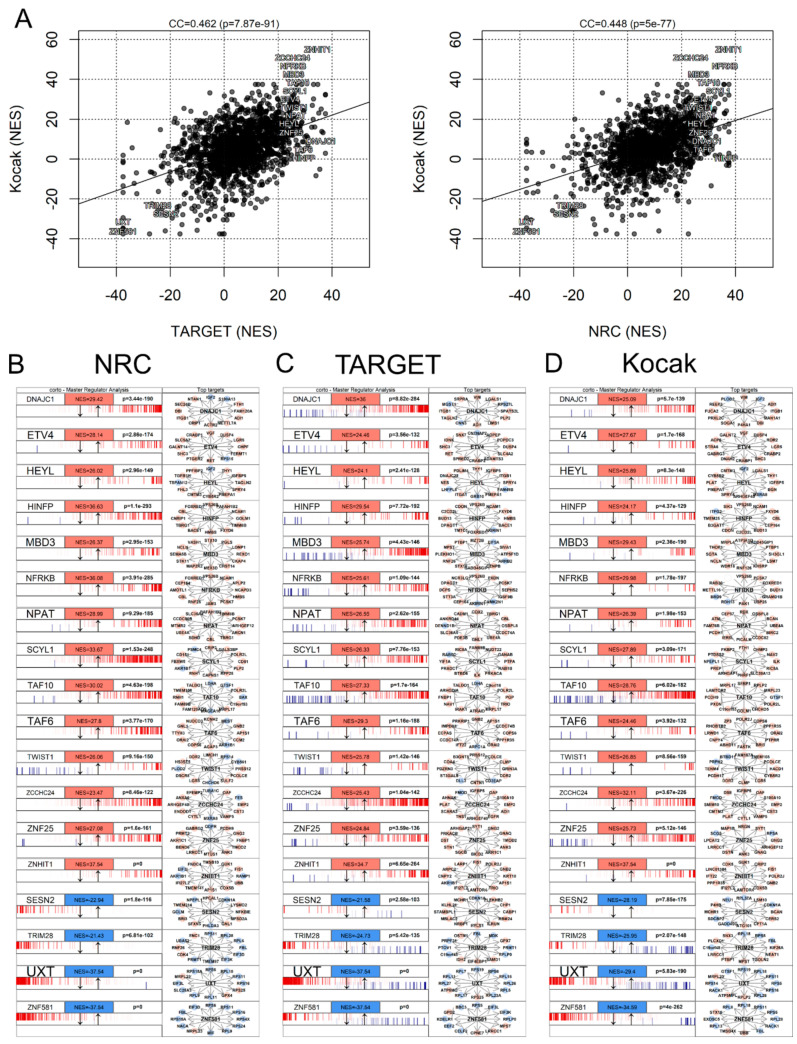
BE2C vs. Kelly master regulator analysis (MRA). (**A**) Comparison between MRA scores derived using the Kocak-based network (*y*-axis) [52], and the networks derived from TARGET and NRC datasets (*x*-axis) [22]. Master regulator scores, as plotted and defined by the corto R package [31] and expressed as NES, based on networks derived from (**B**) the Kocak dataset [52], (**C**) the TARGET dataset [22] and (**D**) the NRC dataset [22].

**Figure 7 biomolecules-11-00177-f007:**
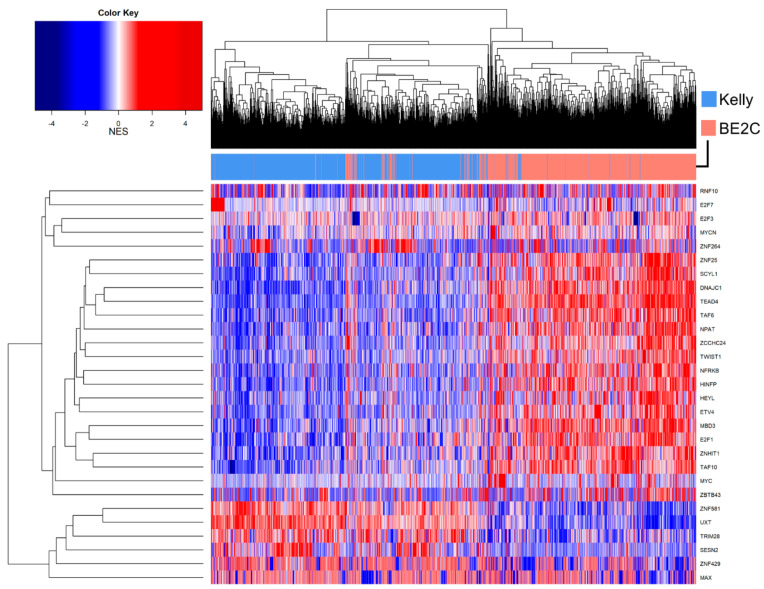
Single-cell master regulator analysis, calculated using the TARGET-derived network [22]. Single-cell scores are reported as NES compared to the mean value of the entire dataset. Cell line is reported on top as salmon (BE2C cells) and cornflower blue (Kelly cells). Cells are clustered using the R hclust algorithm based on Euclidean distance with default parameters.

**Figure 8 biomolecules-11-00177-f008:**
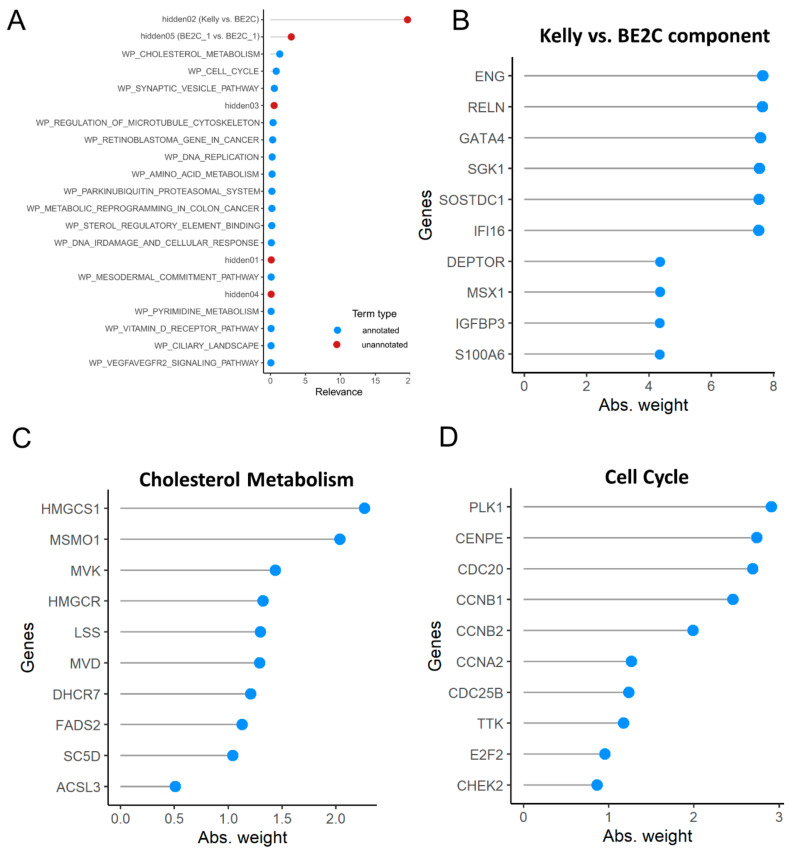
Dissection of heterogeneity in the single-cell dataset, according to the f-scLVM algorithm [69] based on known pathway annotations available from Wiki Pathways [70] and MsigDB [37]. (**A**) Graph showing the most relevant factors identified by the f-scLVM model, both annotated in Wiki Pathways (blue) or not annotated (red). (**B**–**D**) Loadings of the most influential genes in (**B**) “hidden02, Kelly vs. BE2C”, (**C**) “cholesterol metabolism” and (**D**) “cell cycle” terms, defined by the Absolute Weight parameter of the f-scLVM method.

**Table 1 biomolecules-11-00177-t001:** Top 20 marker genes differentiating cluster 2 and cluster 1 of BE2C cells (see Figure 3A), according to Seurat analysis. A negative log fold change indicates lower expression in Cluster 2, and a positive log fold change a higher expression in Cluster 2.

Gene	*p*-Value	Average Log Fold Change	Fraction of Expressing Cells in Cluster 1	Fraction of Expressing Cells in Cluster 2	Adjusted *p*-Value
RPSA	1.30 × 10^−149^	−0.92656	0.998	1	2.05 × 10^−145^
RPL35A	4.43 × 10^−123^	0.482077	1	1	7.00 × 10^−119^
VCAN	7.56 × 10^−123^	−0.74714	0.268	0.962	1.19 × 10^−118^
RPL15	3.18 × 10^−116^	−0.59664	0.998	1	5.01 × 10^−112^
RPL29	3.00 × 10^−115^	−0.41375	1	1	4.73 × 10^−111^
TMA7	5.28 × 10^−111^	−0.53624	0.995	1	8.33 × 10^−107^
SAMD11	2.40 × 10^−108^	−0.72644	0.805	0.99	3.79 × 10^−104^
RPL11	1.11 × 10^−106^	−0.53941	1	1	1.75 × 10^−102^
PPP1R14A	8.54 × 10^−105^	0.899921	0.945	0.428	1.35 × 10^−100^
MAGEA4	2.41 × 10^−104^	0.562087	0.899	0.333	3.80 × 10^−100^
RPL32	4.36 × 10^−102^	−0.43999	1	1	6.88 × 10^−98^
SRM	2.21 × 10^−101^	−0.58422	0.986	1	3.49 × 10^−97^
RPL22	1.89 × 10^−97^	−0.4962	1	1	2.98 × 10^−93^
CDKAL1	2.70 × 10^−96^	−0.64301	0.412	0.933	4.27 × 10^−92^
RPL14	2.26 × 10^−95^	−0.52412	1	1	3.57 × 10^−91^
RPL38	2.68 × 10^−94^	0.437493	1	1	4.23 × 10^−90^
ENO1	6.25 × 10^−90^	−0.53807	0.998	1	9.86 × 10^−86^
RPLP0	1.50 × 10^−89^	−0.30715	1	1	2.37 × 10^−85^
TMEM98	1.62 × 10^−85^	0.543221	0.892	0.474	2.56 × 10^−81^
RPL26L1	2.01 × 10^−84^	0.503347	0.984	0.95	3.17 × 10^−80^

## Data Availability

The raw data associated with this dataset is stored on the National Center for Biotechnology Information (NCBI) and Sequence Read Archive (SRA) servers at https://www.ncbi.nlm.nih.gov/bioproject/?term=PRJNA668226. Data are provided as two BAM files (one per cell line), containing reads aligned on the human GRCh37/hg19 genome and unaligned reads.

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
