# Peer review of "Single-Cell Gene Network Analysis and Transcriptional Landscape of MYCN-Amplified Neuroblastoma Cell Lines"

_biomolecules, 2021, doi:10.3390/biom11020177_

Round 1

Reviewer 1 Report

In this paper the authors report the results of single cell RNA sequencing of ca. 1,000 cells each of two MYCN amplified neuroblastoma cell lines, SK-N-BE-2-C and Kelly. They perform different types of clustering, pathway and gene set enrichment analysis, and master regulator analysis. The results are presented in a descriptive manner without any conclusions other than that there is heterogeneity, which is the result of pretty much any single cell study performed.

As it stands, I do not know what to make of the paper. The number of cell lines and single cell sequences is too small to be a useful resource, and in the absence of conclusions there is little scientific novelty.

Author Response

We have to concur with the reviewer that the first submission of the manuscript was too technical and didn't sufficiently describe and discuss our results. We have expanded significantly the discussion and rewrote large parts of the manuscript to fully vehiculate the biological finding and implications of this analysis. Moreover, we have gone more in-depth in characterizing specific variability and heterogeneity between cells. The dataset is indeed limited to two cell lines, but it provides a high number of samples (2000) and high-coverage (more than 30,000 reads/cells) data, constituting a high quality example of how single cell technology can be applied also to cell line investigation. We believe that this single-cell approach characterizing gene expression, pathway enrichment, and gene network activation in cell cultures is still uncommon and expensive, but it will become more and more popular as technologies become more accessible.

Reviewer 2 Report

Overall comment: The authors obtained single cell RNA sequencing (scRNAseq) data from two cancer cell lines. The single cell transcriptome of the two cell lines are well compared. However, many of the analysis they did is also can be obtained from bulk RNAseq data, which means that their scRNAseq data is not fully analyzed by recently developed bioinformatics analysis pipeline. For example, the differential gene expression analysis between kelly and BE2C can be also done by bulk RNAseq data analysis.

  1. In the abstract and discussion, the authors mentioned "unexpected variability". But it is not clear what is the new finding. I guess it is the high variance of metallothionein (MT2A, MT1X, MT1E) in Kelly cells. It is better to mention it in the abstract.
  2. TSNE and UMAP is not clustering methods. They are dimension reduction and visualization methods. I recommend to use Louvain clustering which is a default algorithm implemented in Seurat package.
  3. The clustering analysis will provide not only the difference between the two cell lines but also the heterogeneity within each cell line. The differentially expressed genes in each sub cluster can be also obtained. Then, I guess that the metallothionein genes are expressed in a sub cluster of Kelly cells. The MYC target expression in Figure 5C can be also captured by clustering analysis. It can be showed in the TSNE, UMAP or heatmap.
  4. The Figure 2E is not a good representation for the comparison between scRNAseq and bulk RNAseq. I recommend to use heatmap of Pearson correlation coefficient between the two scRNAseq data and all the bulkRNAseq data.
  5. In the master regulator analysis, the authors mentioned "highlight the transcription factors". But NOTCH1 is not a transcription factor.
  6. The Figure 6A is not necessary figure to highlight the difference between the two cell line.
  7. NES is only described in the caption of the Figure 5A. It should be described in the methods or results part of the manuscript.

Author Response

Point-by-point answer.

Answer to 1: we thank the reviewer for this very constructive suggestion. We rewrote the abstract, focusing more on results and less on data description, hoping that the peculiarity of a single cell analysis (when compared to bulk RNA-Seq) is now clearer.

Answer to 2: we agree and we apologize for the oversight. We have removed all references to TSNE and UMAP being "clustering" methods, both in the text and in Figure 3. We also applied Louvain clustering (results in Figure 3A, Table 1 and in the Supplementary Materials).

Answer to 3: we thank the reviewer for this suggestion. We expanded the clustering analysis by applying Louvain's method using the Seurat's implementation. Clustering defines two clear groups: Kelly and BE2C. This is unsurprising, as these methods are designed for in vivo single cell data, not for cell cultures, which are intrinsically more uniform than live tissues. However, by playing with the Louvain resolution parameter, we did highlight the presence of two subclusters in BE2C (Figure 3A), characterized by specific markers (Table 1). Whether this is an artifact of the resolution parameter chosen, or a real heterogeneity, we leave the answer to the readers and the data. In the whole paper, we focused on known sources of variance: cell cycle and read coverage. We also highlighted the expression of individual genes (Figure 4), pathways (Figure 5C) and master regulators (Figure 7). We did not use Seurat's functions for plotting, as we used our own code, written in basic and commented R code. We believe that Seurat, while being a great and popular package, and useful for quick pipelines (e.g. the Louvain analysis we performed) is not mandatory for everything single-cell-related, including for example plotting of individual gene expression. Also, the usage of Seurat is limited to predefined pipelines, and wouldn't have allowed us to perform e.g. single cell pathway enrichment or master regulator analysis.

Answer to 4: we thank the reviewer for this suggestion. We have added a Figure 2F besides figure 2E, reporting the correlation coefficients in heatmap form. We also changed the legend for Figure 2E, specifying that it is a TSNE visualization.

Answer to 5: we deeply apologize for this mistake. NOTCH1 has been erroneously included in our list as a transcriptional regulator because Gene Ontology is reporting it as a DNA-binding protein and a regulator of transcription (check here https://www.genecards.org/cgi-bin/carddisp.pl?gene=NOTCH1&keywords=NOTCH1 http://amigo.geneontology.org/amigo/gene_product/UniProtKB:P46531). There will always be an issue of reliance on canonical databases for gene annotation, but thankfully human supervision will always be there to supervise. We thank the reviewer for noticing this. We have manually refined the TF list and corrected the text and figures (6A-D and 7) accordingly, resulting in the removal of NOTCH1 from the Master Regulator Analysis.

Answer to 6: We believe that Figure 6A to be necessary not for comparing BE2C and Kelly (the top results of this comparison are highlighted in figures 6B to 6D) but to highlight the robustness of the Master Regulator Analysis to different network models. That is why we generated networks using three independent large scale NBL datasets. After all, our study not only generates a novel single cell dataset, but also applies for the first time a Master Regulator Analysis on it. Comparing networks is a necessary step of this analysis. We have expanded the text accordingly to specify the purpose of Figure 6A.

Answer to 7: We apologize for the omission. The NES is now described in the methods section.

Round 2

Reviewer 1 Report

The revised version is substantially improved presenting new analyses and putting results into an interpretable context. It is still unclear to me why the authors have performed single cell sequencing, as most of the information also could have been obtained from bulk sequencing. In summary, although I am not enthusiastic about the paper, I find it acceptable in the current form.

Author Response

The revised version is substantially improved presenting new analyses and putting results into an interpretable context. It is still unclear to me why the authors have performed single cell sequencing, as most of the information also could have been obtained from bulk sequencing. In summary, although I am not enthusiastic about the paper, I find it acceptable in the current form.

Response: we thank the reviewer for the positive comments on the improvement of quality and interpretability of our manuscript. Several analyses (e.g. showing the non-uniformity of gene expression, or single-cell master regulator analysis to define TF activity in a cell-by-cell manner) were made possible by the single-cell setup.

Reviewer 2 Report

The authors tried to address my previous comments. However, the revised manuscript still focused on the difference between two cancer cell lines, which can be analyzed by using bulk RNA-sequencing data.

  1. The authors may claim that the pathway analysis and the master regulator analysis would be single-cell-style analysis. However, the resulting list of pathways and master regulators would be able to be obtained by bulk expression data. The authors should show what is the benefit of using single cell RNA sequencing data for cancer cell line analysis. The authors can do pathway and master regulator analysis using bulk expression data and provide them in the Supplementary Material. Then, it would be clearer why single cell analysis is needed to show the difference between two cancer cell lines.
  2. To study the heterogeneity in one cancer cell line, the authors can use some of the single cell analysis tools such as scLVM.
  3. Or the authors also can integrate two cancer cell lines by using CONOS, Harmony, or Seurat Integration. Then the difference between two cancer cell line would be more clear in single cell level.

Author Response

Comment 1: The authors may claim that the pathway analysis and the master regulator analysis would be single-cell-style analysis. However, the resulting list of pathways and master regulators would be able to be obtained by bulk expression data. The authors should show what is the benefit of using single cell RNA sequencing data for cancer cell line analysis. The authors can do pathway and master regulator analysis using bulk expression data and provide them in the Supplementary Material. Then, it would be clearer why single cell analysis is needed to show the difference between two cancer cell lines.

Response to Comment 1: several features of this dataset have been shown at the single-cell level. Single-cell gene expression (Figure 4), single-cell pathway enrichment (Figure 5C) and single-cell master regulator analysis (Figure 7) and gene properties (like number of cells where the gene is expressed, or variance of the gene expression across cells) in Figure 1 C-F. A comparison between single-cell and bulk RNA-Seq is shown in Figure 2C and 2D. These two figures show that bulk and single-cell data are not invalidating each other, but that they are indeed correlated at the most basic level (gene expression quantification). The issue with bulk RNA-Seq data on Kelly and BE2C cells is that it is available, in the reference dataset, only with one replicate. This makes any analysis beyond a quantification of TPMs statistically unsound: for example, master regulator analysis requires, in all the currently accepted implementations, at least 2 replicates per group, in order to test the signature with subsets of the dataset. Nowhere in the paper we claim that bulk RNA-Seq is useless, but we are providing a new single-cell dataset for a much studied pair of cell lines. This allows to draw the same types of results as bulk RNA-Seq, plus also providing new insights on between-cell variability.

Comment 2: To study the heterogeneity in one cancer cell line, the authors can use some of the single cell analysis tools such as scLVM.

Response to Comment 2: This is actually a great suggestion. We did as the reviewer suggested, and applied the most recent implementation of scLVM developed by the same authors of the original method, specifically f-scLVM (factorial single-cell latent variable model). This allowed us to model the sources of heterogeneity in the dataset, which we have included both in the large supplementary document (for reproducibility) and as the novel Figure 8, as well as in the main text. Estimating the heterogeneity of the dataset in this suggested way cannot be performed using a bulk RNA-Seq setup.

Comment 3: Or the authors also can integrate two cancer cell lines by using CONOS, Harmony, or Seurat Integration. Then the difference between two cancer cell line would be more clear in single cell level.

Response to Comment 3: we addressed this point using f-scLVM, as suggested in comment 2.